# Spanish version of the short European Health Literacy Survey Questionnaire HLS-Q12: Transcultural adaptation and psychometric properties

Sergio Muñoz-Villaverde[1,2,3]*, Leticia Serrano-Oviedo[4], María Martínez-García[1,2,5], Yolanda Pardo[6,7,8], Llüisa Tares-Montserrat[1,2], Francisco Javier Gómez-Romero[4,9], Paloma Garcimartin[10,11]

1 Oncology Clinical Trials Unit, Hospital del Mar, Barcelona, Spain, 2 IMIM (Hospital del Mar Medical Research Institute), Cancer Research Program, Barcelona, Spain, 3 Catalan Institute of Health, Casc Antic Primary Care Centre, Barcelona Territorial Management, Barcelona, Spain, 4 Translational Research Unit, University General Hospital of Ciudad Real, Servicio de Saud de Castilla-La Mancha (SESCAM), Spain, 5 Department of Medical Oncology, Hospital del Mar, Parc de Salud Mar, Barcelona, Spain, 6 IMIM (Hospital del Mar Medical Research Institute), Health Services Research Group, Barcelona, Spain, 7 Institute of Health Carlos III, Centre for Biomedical Research Network, Epidemiology and Public Health (CIBERESP), ISCIII, Madrid, Spain, 8 Department of Psychiatry and Legal Medicine, School of Medicine, Universitat Autónoma de Barcelona, Bellaterra, Barcelona, Spain, 9 School of Medicine at Ciudad Real, University of Castilla-La Mancha, Ciudad Real, Spain, 10 Nursing Directorate, Hospital del Mar, Barcelona, Spain, 11 Biomedical Network Research Centre for Cardiovascular Diseases, CIBERCV (Carlos III Health Institute), Madrid, Spain

* sergio.munoz2@alu.uclm.es

## Abstract

### Background

Health literacy has a direct impact on the health of populations. It is related to education, capacity for self-care, and management of health resources. The Health Literacy Survey Questionnaire HLS-Q12 is one of the reference instruments but has not yet been adapted to Spanish. The aims of the study were to cross-culturally adapt and evaluate the psychometric properties of the Spanish version of the HLS-Q12.

### Methods

Data was collected from June 2020 to March 2022. The sample consisted of 60 patients who initiated cancer treatment for the first time within a clinical trial. Double direct translation, back-translation, cognitive debriefing with a 10-patient sample, and an expert committee were used for cross-cultural adaptation. For validation of the HLS-Q12, a psychometric analysis was performed to assess feasibility, reliability, sensitivity to change and construct validity with other measures such as health-related quality of life, empowerment, and health needs.

### Results

The HLS-Q12 is equivalent at the semantic, conceptual, and content level to the original version and its psychometric properties demonstrated good internal consistency with a

**Data Availability Statement:** All relevant data are within the manuscript and its Supporting information files.

**Funding:** The author(s) received no specific funding for this work.

**Competing interests:** The authors have declared that no competing interests exist.

Cronbach's alpha of 0.88 and a McDonald´s omega of 0.91, a high degree of fit for the confirmatory factor analysis, and a statistically significant sensitivity to change (p = 0.025).

## Conclusions

Based on robust psychometric values, the Spanish version of HLS-Q12 was found to be a good cross-culturally adapted tool for collecting correct information on health literacy in cancer patients regardless of tumour type or stage. Although more studies are needed, this version of HLS-Q12 could be used in research for collecting data on the health literacy needs of Spanish-speaking patients.

## Introduction

In recent years health literacy (HL) has received increasing attention from the scientific community. It allows individuals to manage their own health by enhancing self-care capacity, decision-making skills, and active participation in both individualised care and health programmes and therefore has a broad impact on health care systems and public health [1–3]. From this perspective, the World Health Organization (WHO) defines HL as "the social and cognitive skills that determine a person's level of motivation and ability to access, understand and use information in ways that enable them to promote and maintain good health" [4].

In 2015 the European Health Literacy Survey Project (HLS-EU) estimated that 12.4% of the European population had inadequate literacy levels and 35.2% had marginal levels [5]. By contrast, the same study showed that 7.5% of Spanish citizens had inadequate HL levels and 50.8% were marginal [6]. The HLS-EU Project created the HLS-EU-Q86 survey for the measurement of HL. This questionnaire assesses four dimensions of HL (access, understanding, evaluation, and application of health information) that are divided into three different domains (health promotion, disease prevention, and disease care). Later versions such as the HLS-EU-Q47 and others such as the HLS-Q16, HL-SF12, and HLS-Q12 followed [6]. However, the HLS-Q12 does not have an adapted and validated Spanish version [7].

Very few health questionnaires measure HL. The availability of tools that measure HL through easy-to-use and quick-to-administer questionnaires will facilitate the construction of formative education strategies with the aim of increasing patient independence. This is especially true for those with chronic diseases whose personal involvement in the care process is essential to increase the likelihood of adherence to pharmacological therapy and successful disease management [8]. The purpose of this study was to conduct a cross-cultural adaptation and validation of the Spanish shortened version of the HLS-Q12.

## Methods

### Study design and participants

A quantitative psychometric study with a descriptive, longitudinal design and a cross-cultural adaptation was conducted. The study population consisted of patients initiating cancer treatment at the oncology unit of a tertiary hospital in the city of Barcelona. Inclusion criteria were: a) agreement to written informed consent and a follow-up call via telenursing; b) patients >18 years of age; c) patients with solid tumours at any stage of the disease; and d) patients who had not previously received oncological treatment as part of a clinical trial. Exclusion criteria were: a) lack of remote connection devices; b) inability to participate in the study due to clinical

situation (i.e., scores of 2–3 on the Eastern Cooperative Oncology Group (ECOG) performance scale); c) patients with cognitive impairment; d) a level of schooling that did not allow completion of the questionnaires; and e) a language barrier that made it difficult to complete the questionnaires.

For sample size, we considered the number of items that form part of the scale, accepting that the number of subjects for the sample should be between 2 and 10 times the number of items. Based on this, 60 subjects were sufficient for the study [9–14]. Loss-to-follow-up rate of 25% was estimated. Data was collected from June 2020 to March 2022.

## Measurements

Sociodemographic variables included age, gender, marital status, level of education, and whether there were cohabitants at home. Clinical variables included trial phase, tumour type and stage, presence and number of comorbidities and degree of independence. Management of oncological symptomatology was quantified by recording and evaluating expected outcomes and interventions according to the hospital's individualized care plan for cancer patients. CTCAE v5.0 (Common Terminology Criteria for Adverse Events) was used to measure the different degrees of adverse events (G1-G5) related or not to treatment produced by chemotherapeutic, radiotherapeutic, or immunotherapy agents. Among the events collected we highlighted anorexia, nausea, xerostomia, mucositis, diarrhoea, constipation, pain, and fatigue. Pain and fatigue symptoms were measured using the categorical pain scale (no pain-unbearable) and the numerical fatigue scale (0-absence of fatigue, 10-worst possible fatigue), respectively.

HL was measured using the HLS-Q12, a 12-item scale, which measures HL on a Likert scale from 1 (very difficult) to 4 (very easy). The theoretical range is from 12 to 48 points, the higher the score, the higher the HL [7]. The full Spanish HLS-Q12 instrument can be found in S1 Appendix.

Two questionnaires were used to assess the health-related quality of life (HRQoL) of patients: one by the European Organization for Research and Treatment of Cancer (EORTC QLQ-C30) [15,16] and the scale designed by the American Eastern Cooperative Oncology Group (ECOG) [17] and validated by WHO. Empowerment was assessed using the Patient Empowerment in Long-Term Conditions questionnaire (PELC) [9,18] and the Holistic Needs Assessment (HNA) [19,20] to identify individual patient needs.

The 30-item EORTC QLQ-C30 scale incorporates 5 functional dimensions (physical functioning, activities of daily living, emotional functioning, cognitive functioning, and social relationships), 3 symptom scales (fatigue, nausea and vomiting, pain), a global health status scale, and several individual items, to assess additional symptoms commonly reported by cancer patients. All measures range in score from 0 to 100. A high score for the functional scale represents a high/healthy level of functioning, a high score for the global health status represents a high HRQoL, but a high score for a symptom scale represents a high level of symptomatology/problems [15,16].

The ECOG is a hetero-administered scale that assesses the evolution of the patient's abilities in daily life while maintaining maximum autonomy, and its results help to guide therapeutic decisions and the prognosis of the disease. The ECOG is scored from 0 to 5 (normal to death, respectively) [17].

PELC is a self-administered questionnaire that measures empowerment in chronically ill patient and contains 47 items that are scored on a Likert scale from 1 (strongly disagree) to 5 (strongly agree). The scale ranges from 47 to 235, with higher scores indicating higher levels of empowerment [9,18].

The HNA Tool comprises a self-assessment of health needs of patients living with cancer through a simple questionnaire. It measures the physical, practical, emotional, spiritual, social, socio-economic, and environmental need of individuals. A higher number of marked needs indicates a higher number of concerns about the disease process [19,20].

## Data collection

Five visits (V) were made: the first before treatment initiation (V1); 24 hours (V2) and 10 days (V3) after treatment initiation; completion of the educational intervention but before starting a new treatment cycle (V4); and 3 months after enrolment (V5). All patients received an educational intervention with a nurse via synchronous teleconsultation which consisted of information about the clinical trial, resolving doubts about disease process, and providing health education on the warning signs and symptoms of adverse effects of the trial treatment. Both patients and their relatives were informed with the intention of empowering the patient.

Data on sociodemographic and clinical variables were collected at V1. The PELC, EORTC QLQ-C30, HNA, and HLS-Q12 instruments were administered to assess validity and internal consistency of the HLS-Q12. Clinical variables were collected in V2 and V3 and questions were answered based on HNA results. In V4, clinical variables and the PELC, EORTC QLQ-C30, and HNA questionnaires were collected. In V5, information was collected, queries were resolved according to patients' needs, and the treatment status was documented. The self-administered questionnaires were sent by e-mail via Microsoft forms with a mandatory response design. Sociodemographic and clinical data were collected using Research Electronic Data Capture (REDCap). Anonymized study data can be found in S2 Appendix.

## Cross-cultural adaptation process

Permission was received from the author of the original English language version of the HLS-Q12. The aim of the adaptation process is to ensure that the instrument is semantically, conceptually, and content-wise equivalent to the original version. Semantic equivalence seeks to obtain same meaning for each of the items, conceptual equivalence ensures the questionnaire measures same theoretical construct in both cultures, and content equivalence proves each item has relevance for both cultures [18,21]. The adaptation was carried out taking into account the methodology recommended by the International Society for Pharmacoeconomics and Outcomes [22]: 1) forward translation; 2) reconciliation and synthesis of the translations; 3) back translation into English; 4) comparison and harmonization of the back translations with the original; 5) cognitive debriefing; 6) review of the cognitive interviews by the committee of experts; 7) reading test, spelling and grammar check; and 8) drafting of the process report.

## Psychometric properties and data analysis

Sociodemographic and clinical data were collected and expressed as absolute and relative frequencies with means and standard deviations (SDs). The statistical package R software version 4.1.0 was used, and statistically significant values were set to $p < 0.05$.

For psychometric analysis, feasibility, reliability, construct validity, and sensitivity to change were considered [23]. Feasibility was measured by recording the time needed to complete the questionnaire, as well as the descriptive characteristics of the questionnaire items (ceiling and floor effects). To assess relevance of each item, an item-total correlation analysis was performed. It recommended that values $\leq 0.2$ be discarded or reformulated as they represent an insufficient level of homogeneity, items should be at least $\geq 0.3$, and levels $> 0.4$ are considered very good [24].

Reliability was analysed by internal consistency and assessed with Cronbach's alpha and McDonald's omega, both have values between 0–1 with higher values being more reliable [25–28] Cronbach's alpha was also considered if the item was removed; this index shows how reliability improves when the item is removed from the scale, allowing identification of items affecting internal consistency.

For construct validity [29], instrument structure was studied by performing a confirmatory factor analysis (CFA). CFA parameter estimation was carried out using structural equation modelling and the weighted least squares means and variance adjusted (WLSMV) model for calculating estimators as it provided the best option for working with categorical or ordered data [30]. The estimates of latent variables and variances are assumed to take values from -1 to 1 [31]. Absolute fit indicators were considered with chi-square ($\chi^2$), SRMR (standardised root mean residual), and RMSEA (root mean square error of approximation) tests. These indicators are considered adequate for values >0.05 for $\chi^2$, <0.05 for SRMR, and <0.06 for RMSEA [32–34]. Incremental fit indicators utilised were AGFI (Adjusted Goodness of Fit Index), BBNFI (Bentler Bonnet Normed Fit Index), BBNNFI (Bentler Bonnet Non-Normed Fit Index), CFI (Comparative Fit Index), and TLI (Tucker-Lewis Index). These indicators range from 0 (null fit of the model to the data) to 1 (perfect fit); values close to 0.90 are considered adequate [34–36]. Normalised $\chi^2$ was used for parsimony fit indicators which establishes the ratio between $\chi^2$ and the number of degrees of freedom ($\chi^2$/gl). Values <2 are considered acceptable [32–34].

Convergent-divergent validity of the constructs was by performed with a Pearson correlation matrix between HLS-Q12, the rest of the V1 questionnaires, and different clinical variables. The correlation (r) was interpreted as low ($\leq$0.29), moderate (0.30–0.49), and high (>0.50) [37]. For convergent validity, we hypothesised that HLS-Q12 is related to PELC, EORTC-QLQC30 (functioning, symptomatology, and global health), ECOG before intervention, and the number of comorbidities [38–40]. Acceptable convergence is considered when significant correlations p<0.05 were present [26,41]. Regarding divergent validity, we hypothesised that HLS-Q12 has different constructs with HNA, and the clinical variable related to the management of symptomatology in oncological processes [38–40]. Divergence is accepted when a non-significant value p>0.05 is expressed [26,41].

Sensitivity to change was assessed by comparing means of pre-post HLS-Q12 scores, defined by an educational intervention through teleconsultation consisting of five fixed visits plus a series of on-demand consultations (V1 and V4 in the total sample and in two subsamples that we differentiated in the study; stable patients ECOG 0 or experiencing an improvement in ECOG, from 1 to 0, and non-stable patients, who did not show clinical improvement). Cohen's effect size was also calculated. The effect size (ES) values used were considered as either large (>0.8), moderate (0.5), or small (0.2) changes [29,34,42].

## Ethics approval and consent to participate

The project has been approved by the Clinical Research Ethics Committee of the Parc de Salut Mar (No. 2020/9408/I). An information sheet and written informed consent were provided so that participants could be aware of the objectives, the purpose of the study, and could be informed of how their data was managed. At the time of inclusion, they were assigned an identification code consisting of the initials of the recruitment centre and a correlative number.

## Results

### Cross-cultural adaptation of the HLS-Q12

The Spanish HLS-Q12 is presented in Table 1.

**Table 1. Original and adapted HLS-Q12 Health Literacy Questionnaire.**

| On a scale from very difficult to very easy, how easy would you say it is to: | En una escala de "muy difícil" a "muy fácil", indique cuál es el grado de dificultad que encontraría para realizar las siguientes actividades: |
|---|---|
| 1. find information on treatments of illnesses that concern you? | 1. encontrar información sobre los tratamientos de enfermedades que le preocupan? |
| 2. understand what to do in a medical emergency? | 2. entender que debe hacer usted en una emergencia médica? |
| 3. judge the advantages and disadvantages of different treatment options? | 3. valorar las ventajas e inconvenientes de diferentes opciones de tratamientos? |
| 4. follow instructions on medication? | 4. seguir las indicaciones de una medicación? |
| 5. find information on how to manage mental health problems like stress or depression? | 5. encontrar información sobre cómo afrontar problemas de salud mental como el estrés y la depresión? |
| 6. understand why you need health screenings (e.g., breast exam, blood sugar test, blood pressure)? | 6. entender porque es necesario realizarte pruebas médicas (p.ej. mamografía, azúcar en sangre, presión arterial. . .)? |
| 7. judge if the information in the media on health risks is reliable (TV, internet, or other media)? | 7. valorar si la información sobre riesgos para la salud que aparece en los medios de comunicación es fiable (TV, internet u otros)? |
| 8. decide how you can protect yourself from illness based on advice from family and friends? | 8. decidir cómo puede prevenir enfermedades siguiendo consejos de salud de familiares y amigos? |
| 9. find information on healthy activities such as exercise, healthy food and nutrition? | 9. encontrar información sobre actividades saludables como ejercicio, comida sana y nutrición? |
| 10. understand information on food packaging? | 10. entender la información que aparece en los envases de alimentos? |
| 11. judge which everyday behaviour is related to your health (drinking and eating habits, exercise etc.)? | 11. valorar que actividades del día a día influyen sobre su salud (hábitos alimenticios, ejercicio, etc.)? |
| 12. make decisions to improve your health? | 12. tomar decisiones para mejorar su salud? |

A double direct translation of the English questionnaire was completed independently by two bilingual translators whose mother tongue was Spanish and who came from different cultural contexts within the Spanish territory. Both translations were reconciled and synthesised into a single document by a committee of experts yielding a first Spanish version. An expert translator in the healthcare field whose mother tongue was English was contracted to back-translate the document into English. Both English versions were compared, and the synthesised document was sent to the original author who suggested modifications to some of the items, and these opinions were considered by the authors of the Spanish translation who harmonised a second Spanish version.

A cognitive debriefing was conducted via semi-structured and individual interviews with a heterogeneous sample of 10 patients (5 men and 5 women) undergoing cancer treatment who gave their opinion on how the questionnaire was constructed, its ease of use, the type of format, and the brevity and clarity of the questions. 40% of the patients had lung cancers, 40% had genitourinary cancers, 10% had breast cancer, and 10% had digestive cancer. Mean age of the participants was 63.9±10.5 years. A 10% rate of incomprehension was obtained, and participants made suggestions for improvement. Of the 12 questions, two were identified (items 2 and 12) for which participants expressed difficulty in understanding and answering. They were asked for alternative wording and provided ideas to improve their understanding. Of the 10 participants, 30% had marginal, 60% had intermediate, and 10% had advanced levels of HL. The same expert committee reviewed the results of the cognitive interviews, a reading test, and a spelling and grammar check, and then a consensus was reached on the final version of the Spanish questionnaire.

## Psychometric analyses of the HLS-Q12 questionnaire

**Characteristics of the participants.** Table 2 describes the characteristics of the total sample consisting of 60 patients and the two subsamples that were used for the analysis of sensitivity to change, 47 stable patients (78.3%) which ECOG was 0 or experienced an improvement in ECOG, from 1 to 0, and 13 non-stable (21.7%) or worsening patients, who did not show clinical improvement.

**Feasibility.** The average time needed to complete the HLS-Q12 was 4.1±3.9 min (range 1–19 min). Since the questionnaires had an online design with compulsory response, all items were completed for HLS-Q12, QLQC30, PELC, and HNA. In Table 3, it can be observed that scores were distributed over the theoretical range, that the floor effect for the symptomatology dimension of QLQC30 (13.33%), and Dimension (D) 1 (15%), D2 (26.67%), D3 (20%), D4 (45%), and D5 (65%) of HNA obtained the highest percentages and that the questionnaires that obtained the highest ceiling effect percentages were the functioning dimension of the QLQC30 (8.33%), and D3 (5%), D4 (10%), D5 (8.33%), and D6 (5%) of HNA.

**Descriptive analysis of the HLS-12 items.** The frequency of responses for each item, the values of central tendency and dispersion, and the item-total correlation of the HLS-Q12 are shown in Table 4. 75% of items showed a response rate >50% in some of their categories (items 2, 3, 4, 6, 7, 9, 10, 11, 12). All items obtained a correlation >0.51 indicating a high degree of homogeneity and strong association between the item and the total score.

**Reliability.** Cronbach's alpha scores indicated a good internal consistency in HLS-Q12 showing values of 0.88. McDonald´s omega showed values of 0.91.

In addition, Table 5, shows that when the alpha values of some of the items were removed it did not fall below 0.86 indicating a good relationship of the item with the total scale.

**Construct validity.** *Confirmatory factor analysis.* CFA results concluded that HLS-Q12 is a one-dimensional model (Fig 1). Table 6 shows the weight of each of the items in the WLSMV model tested in the study. Most of the items have values >0.4 indicating that all items had a relevant contribution to the model.

*Goodness of fit of the model.* The $\chi^2$ value was significant (p = 0.017), indicating that the hypothesis of a perfect model should be rejected. RMSEA showed a reasonable fit, and although the SRMR did not reach the cut-off value, it was very close. The $\chi^2$/gl ratio and the incremental measure indices (AGFI, BBNFI, BBNNFI, CFI, and TLI) gave acceptable values. The model offers a high degree of fit. A summary can be found in Table 7.

*Convergent-divergent validity.* Pearson's correlation matrix, showed in Table 8, for HLS-Q12 with PELC, EORTC-QLQC30, and HNA and clinical variables indicated that some of the convergent and divergent hypotheses were confirmed.

PELC and the global health dimension of the EORTC-QLQC30 expressed convergence with HLS-Q12. PELC obtained a high correlation with a significant p-value, while the global health dimension of the EORTC-QLQC30 obtained a low correlation, although with a significant p-value. However, functioning and symptom dimensions of EORTC-QLQC30, ECOG before intervention, and number of comorbidities obtained low correlations and non-significant p-values, not expressing convergence with HLS-Q12.

In the divergent hypotheses, low correlations have been established with symptomatology management and HNA, however, the p-value has been significant for HNA and has not expressed significant values with symptomatology management, expressing divergence between symptomatology management and HLS-Q12.

**Sensitivity to change.** Score differences, in Table 9, for HLS-Q12 between the first and fourth visits were statistically significant for the total sample and the subsample of stable or improving patients, but not for the subsample of worsening patients. The ES had a small

**Table 2. Total sample and subsamples of patients selected for psychometric analysis.**

| VARIABLE | | Total sample n = 60 n (%) | Stable sample or improvement n = 47 n (%) | Worsening sample n = 13 n (%) |
|---|---|---|---|---|
| **Gender** | Woman | 20 (33.3) | 14 (29.8) | 7 (53.8) |
| | Male | 40 (66.7) | 33 (70.2) | 6 (46.2) |
| **Age (expressed in min-max.; mean±SD)** | | 45-min 84-max | 45-min 84-max | 45-min 83-max |
| | | 69.1±10.4 | 69.1±10.2 | 69±11.2 |
| **Marital status** | Married | 38 (63.3) | 32 (68.1) | 6 (46.2) |
| | Single | 13 (21.7) | 9 (19.1) | 4 (30.8) |
| | Widowed | 9 (15) | 6 (12.8) | 3 (23.1) |
| **Cohabitation** | Lives accompanied/independent | 38 (63.33) | 32 (68.1) | 7 (53.8) |
| | Lives with someone/carer | 8 (13.33) | 5 (10.6) | 2 (15.4) |
| | Lives alone/independent | 14 (23.33) | 10 (21.3) | 4 (30.8) |
| **Education** | Primary Education | 26 (43.3) | 20 (42.6) | 6 (46.2) |
| | Secondary Education | 26 (43.3) | 21 (44.7) | 5 (38.5) |
| | University Education | 8 (13.3) | 6 (12.8) | 2 (15.4) |
| **Trial phase** | 1 | 13 (21.7) | 8 (17) | 5 (41.7) |
| | 2 | 28 (46.7) | 24 (51.1) | 4 (30.8) |
| | 3 | 19 (31.7) | 15 (31.9) | 4 (30.8) |
| **Tumour type** | Colo-rectal | 6 (10) | 3 (6.4) | 3 (23.1) |
| | Esophagogastric | 2 (3.3) | 2 (4.3) | |
| | Genito-urinary | 27 (45) | 24 (51.1) | 3 (23.1) |
| | Breast | 5 (8.3) | 4 (8.5) | 1 (7.7) |
| | Otorhinolaryngological | 1 (1.7) | 1 (2.1) | |
| | Lung | 18 (30) | 12 (25.5) | 6 (46.2) |
| | Tegumentary | 1 (1.7) | 1 (2.1) | |
| **Stage** | 2 | 11 (18.3) | 8 (17) | 3 (23.1) |
| | 3 | 6 (10) | 5 (10.6) | 1 (7.7) |
| | 4 | 43 (71.7) | 34 (72.3) | 9 (69.2) |
| **Comorbidities** | Yes | 59 (98.3) | 46 (97.9) | 13 (100) |
| | No | 1 (1.7) | 1 (2.1) | 0 (0) |
| **N.º comorbidities** | 0 | 1 (1.7) | 1 (2.1) | |
| | 1 | 4 (6.7) | 4 (8.5) | |
| | 2 | 18 (30) | 16 (34) | 2 (15.4) |
| | 3 | 12 (20) | 8 (17) | 4 (30.8) |
| | 4 | 9 (15) | 7 (14.9) | 2 (15.4) |
| | 5 | 8 (13.3) | 7 (14.9) | 1 (7.7) |
| | 6 | 5 (8.3) | 3 (6.4) | 2 (15.4) |
| | 8 | 2 (3.3) | | 2 (15.4) |
| | 11 | 1 (1.7) | 1 (2.1) | |
| **ECOG before intervention** | 0 | 37 (61.7) | 31 (66) | 6 (46.2) |
| | 1 | 23 (38.3) | 16 (34) | 7 (53.8) |
| **ECOG after intervention** | 0 | 35 (59.3) | 35 (74.5) | |
| | 1 | 17 (28.8) | 12 (25.5) | 5 (41.7) |
| | 2 | 2 (3.4) | | 2 (16.7) |
| | 3 | 5 (8.5) | | 5 (41.7) |
| **Symptomatology management** | Yes | 52 (86.7) | 44 (93.6) | 8 (61.5) |
| | No | 8 (13.3) | 3 (6.4) | 5 (38.5) |

ECOG: Eastern Cooperative Oncology Group; SD: Standard deviation.

**Table 3. Scores and feasibility coefficients of the QLQ-C30, PELC, HNA, and HLS-Q12 questionnaires (n = 60).**

| Questionnaires | Theoretical range | Items with MV, % | Feasibility | | | | |
|---|---|---|---|---|---|---|---|
| | | | Observe range | Mean | SD | Floor % | Ceiling% |
| **EORTC QLQ-C30** | | | | | | | |
| • Functional scales | 0–100 | 0 | 10.7–100 | 72.2 | 22.9 | 1.67 | 8.33 |
| • Symptom scales | 0–100 | 0 | -1.2–74.7 | 21.6 | 18.3 | 13.33 | 1.67 |
| • Global health status | 0–100 | 0 | 0–100 | 54.2 | 22.9 | 1.67 | 3.33 |
| **PELC** | 47–235 | 0 | 109–221 | 161.2 | 22.1 | 1.67 | 1.67 |
| **HNA** | | | | | | | |
| • D1+Physical concerns | 0–28 | 0 | 0–28 | 9.9 | 7.6 | 15 | 1.67 |
| • D2+Practical concerns | 0–16 | 0 | 0–16 | 3.6 | 3.9 | 26.67 | 1.67 |
| • D3+Emotional concerns | 0–12 | 0 | 0–12 | 4.2 | 3.5 | 20 | 5 |
| • D4+Family/relationship concerns | 0–5 | 0 | 0–5 | 1.3 | 1.6 | 45 | 10 |
| • D5+Spiritual or religious concerns | 0–3 | 0 | 0–2 | 0.4 | 0.6 | 65 | 8.33 |
| • D6+Lifestyle or information needs | 0–11 | 0 | 0–11 | 4.3 | 3 | 13.33 | 5 |
| **TOTAL** | 0–75 | 0 | 0–69 | 23.9 | 17.2 | 11.67 | 1.67 |
| **HLS-Q12** | 12–48 | 0 | 21–45 | 32.4 | 5.2 | 1.67 | 1.67 |

D: Dimension; EORTC QLQ-C30: European Organisation for Research and Treatment of Cancer Quality of Life Questionnaire—Core 30; HLS-Q12: Health Literacy Survey Questionnaire; HNA: Health Needs Assessment tool; MV: missing values; PELC: Patient Empowerment in Long-Term Conditions questionnaire; SD: Standard deviation.

**Table 4. Descriptive analysis of the HLS-Q12.**

| Item content | MV n (%) | Very difficult n (%) | Difficult n (%) | Easy n (%) | Very easy n (%) | Item-total correlation |
|---|---|---|---|---|---|---|
| Item 1 | 0 (0) | 6 (10) | 28 (46.7) | 20 (33.3) | 6 (10) | 0.73 |
| Item 2 | 0 (0) | 4 (6.7) | 16 (26.7) | 38 (63.3) | 2 (3.3) | 0.62 |
| Item 3 | 0 (0) | 3 (5) | 32 (53.3) | 23 (38.3) | 2 (3.3) | 0.58 |
| Item 4 | 0 (0) | 1 (1.7) | 8 (13.3) | 43 (71.7) | 8 (13.3) | 0.65 |
| Item 5 | 0 (0) | 5 (8.3) | 23 (38.3) | 28 (46.7) | 4 (6.7) | 0.72 |
| Item 6 | 0 (0) | 0 (0) | 5 (8.3) | 38 (63.3) | 17 (28.3) | 0.56 |
| Item 7 | 0 (0) | 2 (3.3) | 35 (58.3) | 20 (33.3) | 3 (5) | 0.51 |
| Item 8 | 0 (0) | 4 (6.7) | 29 (48.3) | 22 (36.7) | 5 (8.3) | 0.57 |
| Item 9 | 0 (0) | 0 (0) | 16 (26.7) | 37 (61.7) | 7 (11.7) | 0.75 |
| Item 10 | 0 (0) | 2 (3.3) | 17 (28.3) | 36 (60) | 5 (8.3) | 0.71 |
| Item 11 | 0 (0) | 1 (1.7) | 8 (13.3) | 43 (71.7) | 8 (13.3) | 0.74 |
| Item 12 | 0 (0) | 0 (0) | 15 (25) | 38 (63.3) | 7 (11.7) | 0.76 |

MV: Missing values.

coefficient for the total sample and the subsample of stable or improving patients, but a large effect for the subsample of worsening patients.

## Discussion

This study describes the process of cross-cultural adaptation for HLS-Q12 and its psychometric properties in terms of feasibility, reliability, construct validity, and sensitivity to change. The results indicate that the Spanish translation is equivalent at the semantic, conceptual, and

**Table 5. Cronbach´s alpha if item is removed.**

| Item content | Cronbach´s alpha if item is removed |
|---|---|
| Item 1 | 0.87 |
| Item 2 | 0.87 |
| Item 3 | 0.87 |
| Item 4 | 0.87 |
| Item 5 | 0.87 |
| Item 6 | 0.87 |
| Item 7 | 0.88 |
| Item 8 | 0.88 |
| Item 9 | 0.86 |
| Item 10 | 0.86 |
| Item 11 | 0.86 |
| Item 12 | 0.86 |

content levels to the original version with good internal consistency, and a small-moderate ES, but statistically significant sensitivity to change. The construct validity showed mixed results as factor analysis verified the structure of the original model with a high degree of fit, but not all convergent-divergent validity hypotheses were confirmed.

According to the HLS-EU project, it is necessary to have reduced instruments adapted and validated in different languages that are customised to particularities and contexts where they are applied. HLS-EU-Q86 was followed by reduced versions including HLS-EU-Q47, HLS-Q16, HL-SF12, and HLS-Q12 [6]. For example, HLS-Q47 in an Asian sample that showed good construct validity, satisfactory goodness-of-fit with a 3-domain model, high internal consistency, adequate convergent validity, and no significant ceiling/floor effects [43]. For HLS-Q12, validation results showed good construct validity, adequate goodness-of-fit with a 3-domain model, and high internal consistency [7,44].

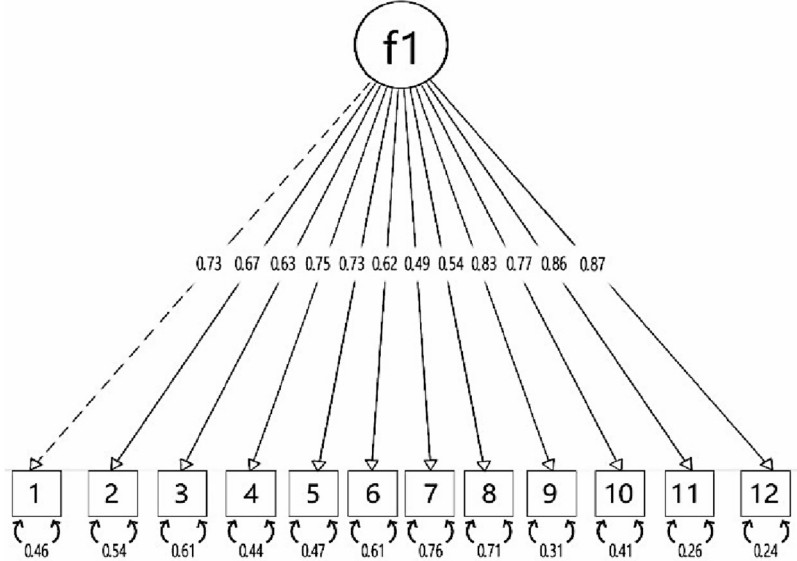

**Fig 1. Confirmatory Factor Analysis model for the HLS-Q12.** f1, single factor model.

**Table 6. Estimated parameters of the confirmatory factor analysis.**

| Variable | Standardised loading factor* | Standardised error variances* |
|---|---|---|
| Item 1 | 0.734 | 0.461 |
| Item 2 | 0.675 | 0.545 |
| Item 3 | 0.625 | 0.609 |
| Item 4 | 0.751 | 0.436 |
| Item 5 | 0.729 | 0.468 |
| Item 6 | 0.621 | 0.615 |
| Item 7 | 0.489 | 0.761 |
| Item 8 | 0.542 | 0.707 |
| Item 9 | 0.832 | 0.307 |
| Item 10 | 0.769 | 0.409 |
| Item 11 | 0.863 | 0.255 |
| Item 12 | 0.874 | 0.235 |

*Standardised factor loadings.

In our translation of HLS-Q12, feasibility showed a good response rate due to the online, compulsory response design, together with a reduced number of questions. The item-total correlation of the HLS-Q12 obtained values >0.50 suggesting a strong homogeneity and that each item measures the same construct [24]. Values >0.7 were obtained for reliability scores both for the total sum of the instrument (alfa and omega) and for the individual items, which are considered acceptable to indicate that an instrument has good internal consistency [25–28]. These results agree with those obtained by previous versions, with values of 0.87 for HL-SF12, 0.85 for HLS-Q12 in its English version, 0.98 for HLS-Q16 in its Spanish version, and values of >0.90 in its original version HLS-EU-Q47 [43–45].

For construct validity, the original instrument is described as a unidimensional model, which also corroborates results of this study [7]. However, another study using a different methodology (Rasch model) described HLS-Q12 as having a three-subdimensional model [44]. In our study CFA showed values for $X^2$/gl, SRMR, RMSEA, CFI, TLI, BBNNFI, AGFI, and BBNFI, that are in general better than those of the original HLS-EU-Q47 [43].

**Table 7. Goodness-of-fit indicators of the confirmatory factor analysis.**

| Index | Acceptable level of fit | Obtained value | |
|---|---|---|---|
| **Goodness-of-fit test** | p>0.05 | $\chi^2$ = 78,457<br>gl = 54<br>p = 0.017 | Unacceptable |
| **Chi² normalised** | $\chi^2$/gl<2 | $\chi^2$/gl = 1.45 | Acceptable |
| **SRMR** | <0.05 | 0.11 | Unacceptable |
| **RMSEA** | <0.06 | 0.08 | Reasonable adjustment |
| **CFI** | >0.9 | 0.98 | Acceptable |
| **TLI** | >0.9 | 0.98 | Acceptable |
| **AGFI** | >0.9 | 0.95 | Acceptable |
| **BBNFI** | >0.9 | 0.95 | Acceptable |
| **BBNNFI** | >0.9 | 0.98 | Acceptable |

AGFI: Adjusted Goodness of Fit Index; BBNFI: Bentler Bonnet Normed Fit Index; BBNNFI: Bentler Bonnet Non-Normed Fit Index; CFI: Comparative Fit Index; gl: degrees of freedom; RMSEA: Root mean square error of approximation; SRMR: Standardised root mean residual; TLI: Tucker-Lewis Index; $\chi^2$: chi-square.

**Table 8. Pearson multitrait-multimethod correlation matrix for the assessment of the validity of HLSQ12.**

| | HLS-Q12 | |
| --- | --- | --- |
| | **Convergent validity** | **Divergent validity** |
| **PELC** | *r* = 0.53 <br> *related p<0.001 | |
| **EORTC-QLQC30** | | |
| • **Functional scales** | *r* = 0.12 <br> unrelated p = 0.29 | |
| • **Symptom scales** | *r* = -0.12 <br> unrelated p = 0.24 | |
| • **Global health status** | *r* = 0.19 <br> *related p = 0.022 | |
| **HNA** | | *r* = -0.16 <br> related p = 0.046 |
| **ECOG before intervention** | *r* = -0.15 <br> unrelated p = 0.099 | |
| **Number of comorbidities** | *r* = -0.08 <br> unrelated p = 0.31 | |
| **Symptomatology management** | | *r* = -0.08 <br> *unrelated p = 0.67 |

ECOG: Eastern Cooperative Oncology Group; EORTC-QLQ-C30: EORTC Quality of Life Questionnaire; HLS-Q12: Health Literacy Survey Questionnaire; HNA: Health Needs Assessments tool; PELC: Patient Empowerment in Long-Term Conditions questionnaire.

*Confirmed hypothesis.

Regarding convergent-divergent validity, based on findings in the literature we were challenged to define hypotheses of interrelationships between literacy and other concepts as well as with other health questionnaires and different clinical variables [38–40]. For convergent validity, the hypotheses we proposed regarding the relationship between HL with PELC and the global health dimension of the EORTC-QLQC30 were fulfilled, indicating that the higher the literacy level, the greater the empowerment and the more positive the perception of health status, which is in agreement with the literature [38–40]. We observed a positive correlation between literacy level and how it influences better patient self-care and a better perception of global health. Conversely, we could not confirm the convergence hypotheses with EORTC-QLQC30 constructs of functioning and symptomatology, ECOG before intervention, and the number of comorbidities. Some previous studies have shown that the higher HL, the better the quality of life and the lower the reported symptomatology, and that the number of previous comorbidities negatively affects the HL of individuals [38–40]. We could not establish

**Table 9. Estimates of sensitivity to change in the total sample and subsamples for HLS-Q12.**

| | Total sample (n = 60) | | | Stable or improving subsample (n = 47) | | | Worsening subsample (n = 13) | | |
| --- | --- | --- | --- | --- | --- | --- | --- | --- | --- |
| | **Change (mean±SD)** | **p-value*** | **ES** | **Change (mean±SD)** | **p-value*** | **ES** | **Change (mean±SD)** | **p-value*** | **ES** |
| **HLS-Q12** | 3.3±14.5 | 0.025 | 0.31 | 0.30±12 | 0.047 | 0.03 | 14±17.8 | 0.158 | 1.07 |

Comparing mean change of pre-post intervention in HLS-Q12 scores in total sample and subsamples; stable patients ECOG 0 or experiencing an improvement in ECOG, from 1 to 0, and worsening patients.

ECOG: Eastern Cooperative Oncology Group; ES: effect size; HLS-Q12: Health Literacy Survey Questionnaire; SD: Standard deviation.

* The p-values correspond to the result of the t-test (parametric) calculation after checking normality of the comparing mean changes of pre-post intervention in HLS-Q12 scores.

these hypotheses, which aligns with previous findings that show that correlations do not occur in all cases [38–40]. That some of the relationships were not significant may result from our sample size, but even so the directions of the correlations are in the direction we hypothesised.

Findings on confirmation of the divergent validity of HLS-Q12 establishes that HL alone does not have a positive relationship on the correct management of cancer symptomatology [38]. This finding could be due to the complex aetiology of cancer where symptomatology can affect each person differently depending on the stage or severity of the cancer disease, their quality of life, and their psychological state [38–40,46]. The HNA tool indicated that, for HLS-Q12, higher level of HL correlated with higher number of health needs expressed [38]. Our results showed that HL equated to lower number of expressed needs, therefore we reject our divergent hypothesis. The literature is ambiguous in this area as data are found where, despite good HL, patients continue to express doubts and changing needs [38,39]. As with convergence, sample size could have an influence.

For sensitivity to change, both overall and the stable or improvement subsamples showed a small ES but with a statistically significant p-values. This may indicate that, although the changes in scores were small, there was an increase in patient literacy. In comparison, the subsample of worsening patients had a large ES and a non-significant p-value. This could indicate that in the subsample of patients where the ECOG was worse the educational intervention achieved a substantial change in HL level since their previous literacy levels were lower, however, the sample of patients is too small to obtain a significant p-value. None of the studies concerning the HLS questionnaires have considered sensitivity to change, which limits result comparisons [43–45].

## Limitations

The main limitation of the validation study is the sample size. Although the number of participants should be 2 to 10 times the number of questions contained in the questionnaire [13], others have suggested the minimum should be 5 participants per item [14]. In our case, for a 12-question survey with 60 participants, a participant/question ratio of 5 could be considered adequate but a minimum of 200 participants would be optimal for high communalities and well-determined factors and for low communalities and poorly determined factors this number could be as high as 500 [47,48]. The most penalizing aspect of not having a larger sample was in the formation of the subsamples of sensitivity to change. However, ECOG is widely used in oncology to describe patient's quality of life and self-care and degree of independence, which is a good indicator of patient stability [17].

For reliability, we did not consider calculating the Intraclass Correlation Coefficient (ICC), which measures the temporal stability of the responses to the questionnaire in the same patient in the same conditions [26,29], given that we proposed an educational intervention from the first visit.

Follow-up times for educational interventions were variable and dependent upon the clinical trial which could impact HL measurement. Even so, there is a lack of consensus when it comes to defining exact times that correctly identify sensitivity to change [42].

## Strengths and future studies

To the best of our knowledge, this is the first study to validate the HLS-Q12 in the Spanish context and in a heterogeneous oncologic population. The original questionnaires were developed in large samples of patients coming from different health backgrounds, however, there are very few instruments that address the oncological setting [49–51]. The fact that this adaptation and validation of the HLS-Q12 in Spanish has exclusively been conducted in an oncology

population may be considered a strength in the absence of previous studies of the HLS project in cancer patients in Spain.

This is one of the first studies to perform convergent-divergent validity with several health questionnaires and different clinical variables [43,45], although studies with larger samples would still be necessary to improve these results and for the comparison of data in terms of sensitivity to change.

This tool could serve as an instrument for conducting multicentre studies that would provide information on the HL of patient populations in Spain, which could facilitate the development of effective and efficient strategies aimed at improving self-care, control, and critical and contrasted decision-making by patients about their own disease [8].

## Conclusions

The results of our study suggest that HLS-Q12 is a robust cross-culturally adapted tool for collecting accurate HL information in cancer patients regardless of tumour or stage. More studies are needed to confirm and expand these findings, but our results support that HLS-Q12 could be used to enable interventions aimed at reducing gaps in health services and even influence the discussion of policy strategies to help improve equality and equity in health, especially in the field of oncology.

## Supporting information

**S1 Appendix. Health Literacy Survey-Questionnaire 12 (Spanish version).**
(PDF)

**S2 Appendix. Anonymized study data.**
(XLSX)

## Acknowledgments

The authors would like to thank Hanne Søberg Finbråten, Guillermo Pedreira Robles, Juan Riesgo Martín, and Ágata Aguirre Martínez for their contributions.

Editorial assistance, in the form of language editing and correction, was provided by XpertScientific Editing and Consulting Services.

## Author Contributions

**Conceptualization:** Sergio Muñoz-Villaverde, Leticia Serrano-Oviedo, María Martínez-García, Paloma Garcimartin.

**Data curation:** Sergio Muñoz-Villaverde, Leticia Serrano-Oviedo, María Martínez-García, Yolanda Pardo, Paloma Garcimartin.

**Formal analysis:** Sergio Muñoz-Villaverde, Yolanda Pardo, Francisco Javier Gómez-Romero.

**Investigation:** Sergio Muñoz-Villaverde, Leticia Serrano-Oviedo, María Martínez-García, Yolanda Pardo, Llüisa Tares-Montserrat, Paloma Garcimartin.

**Methodology:** Sergio Muñoz-Villaverde, Leticia Serrano-Oviedo, María Martínez-García, Yolanda Pardo, Paloma Garcimartin.

**Project administration:** Sergio Muñoz-Villaverde, Leticia Serrano-Oviedo, María Martínez-García, Paloma Garcimartin.

**Resources:** Sergio Muñoz-Villaverde, María Martínez-García, Llüisa Tares-Montserrat, Paloma Garcimartin.

**Supervision:** Sergio Muñoz-Villaverde, Leticia Serrano-Oviedo, María Martínez-García, Paloma Garcimartin.

**Validation:** Sergio Muñoz-Villaverde, Leticia Serrano-Oviedo, María Martínez-García, Yolanda Pardo, Llüisa Tares-Montserrat, Francisco Javier Gómez-Romero, Paloma Garcimartin.

**Visualization:** Sergio Muñoz-Villaverde, Leticia Serrano-Oviedo, María Martínez-García, Yolanda Pardo, Paloma Garcimartin.

**Writing – original draft:** Sergio Muñoz-Villaverde.

**Writing – review & editing:** Sergio Muñoz-Villaverde, Leticia Serrano-Oviedo, María Martínez-García, Yolanda Pardo, Llüisa Tares-Montserrat, Francisco Javier Gómez-Romero, Paloma Garcimartin.

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
