## [Decision Letter · Decision Letter 0]

29 Nov 2023

PONE-D-22-34462Spanish version of the short European Health Literacy Survey Questionnaire HLS-Q12: transcultural adaptation and psychometric propertiesPLOS ONE

Dear Dr. Muñoz-Villaverde,

Thank you for submitting your manuscript to PLOS ONE. After careful consideration, we feel that it has merit but does not fully meet PLOS ONE’s publication criteria as it currently stands. Therefore, we invite you to submit a revised version of the manuscript that addresses the points raised during the review process.

We look forward to receiving your revised manuscript.

Kind regards,

I Gede Nyoman Mindra Jaya

Academic Editor

PLOS ONE

Journal Requirements:

Reviewers' comments:

Reviewer's Responses to Questions

**Comments to the Author**

1. Is the manuscript technically sound, and do the data support the conclusions?

Reviewer #1: Yes

Reviewer #2: Yes

2. Has the statistical analysis been performed appropriately and rigorously? 

Reviewer #1: Yes

Reviewer #2: Yes

3. Have the authors made all data underlying the findings in their manuscript fully available?

Reviewer #1: Yes

Reviewer #2: Yes

4. Is the manuscript presented in an intelligible fashion and written in standard English?

Reviewer #1: Yes

Reviewer #2: Yes

5. Review Comments to the Author

Reviewer #1: The paper is well written. Nevertheless, here are my concern to improve the paper.

1. Since the authors had already used CFA, I suggest the authors to replace the reliability analysis by the one based on CFA. The authors may refer to Hair et al. (2014; p. 619) for construct reliability and Bollen (1989; p. 221) for item reliability.

In Tabel 6:

2. The column “Predictor” may be deleted.

3. I suggest to change “Standardised latent variable” to “standardised loading factor” and “Standardised variances” to “Standardised error variances”.

In Table 9:

4. the title “Estimators”, change to “Estimates”.

5. Please clarify the statistical test relating to the “p-value”!

Reviewer #2: The article highlights and justifies the need to adapt and validate the questionnaire European Health Literacy Survey Questionnaire HLSQ12 to Spanish.

One of the possible important biases is the lack of sample, but the authors correctly justify the limitations and propose how to correct it in future research.

In the description of the groups, although it is specified in Table 3, it would provide the most relevant data in the text with the total samples, as it has done with one of the subsamples.

The bibliographic references are well-structured, relevant and up-to-date.

Regarding the form, with reference to the final questionnaire and Table 1, homogenize the use of capital letters in all questions.

6. PLOS authors have the option to publish the peer review history of their article (what does this mean?). If published, this will include your full peer review and any attached files.

Reviewer #1: No

Reviewer #2: No

---

## [Author Response · Author response to Decision Letter 0]

27 Dec 2023

Response to reviewers and editor

Thank you for your kind contributions. We have introduced changes based on the suggestions that will contribute to enrich our manuscript. The following is a point-by-point reply to each of the considerations that we received.

Editor- When submitting your revision, we need you to address these additional requirements.

Thank you for reminding us to double check PLOS ONE´s style requirements. We have done this and have addressed accordingly as follows.

1. Please ensure that your manuscript meets PLOS ONE's style requirements, including those for file naming. The PLOS ONE style templates can be found at https://journals.plos.org/plosone/s/file?id=wjVg/PLOSOne_formatting_sample_main_body.pdf andhttps://journals.plos.org/plosone/s/file?id=ba62/PLOSOne_formatting_sample_title_authors_affiliations.pdf

After a meticulously read, we have followed PLOS ONE´s style requirements and highlighted accordingly throughout the text, so changes can be reflected. We hope that no requirements have been missed.

In methods and in the ethics approval and consent to participate sections, the word written has been added in lines 96, 240 to informed consent to state the type of consent that was obtained.

3. Please review your reference list to ensure that it is complete and correct. If you have cited papers that have been retracted, please include the rationale for doing so in the manuscript text or remove these references and replace them with relevant current references. Any changes to the reference list should be mentioned in the rebuttal letter that accompanies your revised manuscript. If you need to cite a retracted article, indicate the article’s retracted status in the References list and also include a citation and full reference for the retraction notice.

The reference list has been checked and ensure that to our knowledge it is correct. Dois have been added to some of the references, these have been highlighted in the reference section within the manuscript.

Reviewer #1. Reviewer 1 stated:

The paper is well written. Nevertheless, here are my concern to improve the paper.

Thank you so much for your feedback and appreciation. About the concerns to improve the paper:

Comment #1. Since the authors had already used CFA, I suggest the authors to replace the reliability analysis by the one based on CFA. The authors may refer to Hair et al. (2014; p. 619) for construct reliability and Bollen (1989; p. 221) for item reliability.

Thank you for your advice and for sharing those references, to keep up to date with new recommendations for internal consistency measuring we have considered your references incorporating McDonald´s omega (construct reliability/composite reliability) calculated through SEM. However, we have considered to leave alpha´s value as it is the traditional of way reporting reliability and yet is still considered a valid measurement of reliability. Hope this is not a problem.

We have incorporated up-to-date references accordingly lines 588,591. We have also introduced lines 56, 200, 310, 397 in abstract, methods, results and in discussion to incorporate McDonald´s omega.

In Table 6: Comment #2. The column “Predictor” may be deleted.

Comment #3. I suggest to change “Standardised latent variable” to “standardised loading factor” and “Standardised variances” to “Standardised error variances”.

Thank you for these suggestions. As per your recommendation this column has been deleted as it does not add any extra value to the table. And in the manuscript in line 320 it is already stated that HLS-Q12 is a one-dimensional model.

Also, we have taken into account your suggestion a have renamed Standardised latent variable” to “Standardised loading factor” and “Standardised variances” to “Standardised error variances” in table 6 in line 324.

In Table 9: Comment #4. the title “Estimators”, change to “Estimates”.

As recommended Estimators have been changed to Estimates in table 9 in line 367.

In table 9: Comment #5. Please clarify the statistical test relating to the “p-value”!

Thank you for indicating clarification for the statistical test use relating the p-value. Due to normality of the variables, t-test was used for calculation of p-values in the total sample and subsamples. The following comment has been added to the table on the calculation of p-values for a better understanding in lines 372,373: * The p-values correspond to the result of the t-test (parametric) calculation after checking normality of the comparing mean changes of pre-post intervention in HLS-Q12 scores.

Reviewer #2. Reviewer 2 stated:

Comment #1. The article highlights and justifies the need to adapt and validate the questionnaire European Health Literacy Survey Questionnaire HLSQ12 to Spanish. One of the possible important biases is the lack of sample, but the authors correctly justify the limitations and propose how to correct it in future research.

Thank you for comment. As you have correctly highlighted, this has been accordingly justified in the limitation’s sections, however the fact that this questionnaire has been validated in an oncology population brings an important value to the scientific community, as this may be considered a strength in the absence of previous studies of the HLS project in cancer patients in Spain.

Comment #2. In the description of the groups, although it is specified in Table 3, it would provide the most relevant data in the text with the total samples, as it has done with one of the subsamples.

Thank you so much for your input regarding the description of the groups. In the text before the table that describes the total sample and subsamples, the following information has been added in lines 276-279 to expand the information regarding the samples:

“describes the characteristics of the total sample consisting of 60 patients and the two subsamples that were used for the analysis of sensitivity to change, 47 stable patients (78.3%) which ECOG was 0 or experienced an improvement in ECOG, from 1 to 0, and 13 non-stable (21.7%) or worsening patients, who did not show clinical improvement”.

Comment #3. The bibliographic references are well-structured, relevant and up-to-date.

This is appreciated, thank you so much for your input. However, some updates have been made to references as a couple of references needed to be incorporated for a better justification. These have been highlighted in the reference section within the manuscript. Also, Dois have been incorporated into the references.

Comment #4. Regarding the form, with reference to the final questionnaire and Table 1, homogenize the use of capital letters in all questions.

Thank you for your comment, however we have wanted to follow the same structure of the original questionnaire, and hence the questions are kept in low cases. In addition, in S1 Appendix (final questionnaire), capital letter F has been added for grade number 3. Fácil, as in the previous appendix this was in low cases.

After a complete and a thorough review of the manuscript, and adding to the comments from the reviewers, we the authors have found out a transcription error in the theoretical range of the PELC questionnaire, which is 47-235 and not 63-219 as stated in the manuscript. We do apologize for this data error and have updated through the text in line 149. And in table 3, line 294-295.

Also, maximum likelihood estimation in line 193 in the original manuscript was deleted, as the model that we considered for construct validity and CFA was the the weighted least squares means and variance adjusted (WLSMV) model as it was already stated in lines 193-194 in the original manuscript and in line 207 in the revised manuscript.

---

## [Editor Report · Decision Letter 1]

16 Feb 2024

Spanish version of the short European Health Literacy Survey Questionnaire HLS-Q12: transcultural adaptation and psychometric properties

PONE-D-22-34462R1

Dear Dr. Muñoz-Villaverde,

We’re pleased to inform you that your manuscript has been judged scientifically suitable for publication and will be formally accepted for publication once it meets all outstanding technical requirements.

Kind regards,

I Gede Nyoman Mindra Jaya

Academic Editor

PLOS ONE
---

## [Editor Report · Acceptance letter]

19 Feb 2024

PONE-D-22-34462R1 

PLOS ONE

Dear Dr. Muñoz-Villaverde, 

I'm pleased to inform you that your manuscript has been deemed suitable for publication in PLOS ONE. Congratulations! Your manuscript is now being handed over to our production team.

Kind regards, 

on behalf of

Dr. I Gede Nyoman Mindra Jaya 

Academic Editor

PLOS ONE